# Frequent Users in Psychiatric Consultations: A 6-Year Retrospective Study in the Emergency Department

**DOI:** 10.3390/ijerph22060828

**Published:** 2025-05-23

**Authors:** Carla Maria Gramaglia, Eleonora Gambaro, Alessandro Feggi, Amalia Jona, Valentina Zanoli, Francesco Gavelli, Gian Carlo Avanzi, Daniela Ferrante, Silviana Maria Patratanu, Erica Valerio, Patrizia Zeppegno

**Affiliations:** 1Department of Translational Medicine, University of Eastern Piedmont, Via Solaroli, 17, 28100 Novara, Italy; eleonora.gambaro@uniupo.it (E.G.); alessandro.feggi@maggioreosp.novara.it (A.F.); patrizia.zeppegno@med.uniupo.it (P.Z.); 2Psychiatry Unit Maggiore della Carità University Hospital, Corso Mazzini, 18, 28100 Novara, Italy; amalia.jona@maggioreosp.novara.it (A.J.); valentina.zanoli@maggioreosp.novara.it (V.Z.); 3Emergency Medicine Unit, Maggiore della Carità University Hospital, Corso Mazzini, 18, 28100 Novara, Italy; francesco.gavelli@med.uniupo.it (F.G.); giancarlo.avanzi@uniupo.it (G.C.A.); 4Unit of Medical Statistics, Department of Translational Medicine, University of Eastern Piedmont, Via Solaroli, 17, 28100 Novara, Italy; daniela.ferrante@uniupo.it; 5University of Eastern Piedmont, Via Ettore Perrone, 18, 28100 Novara, Italy; silvianamaria.patratanu@gmail.com (S.M.P.); 20051737@studenti.uniupo.it (E.V.)

**Keywords:** emergency service, hospital, frequent users, health services misuse, mental health, socioeconomic factors, demography, psychiatric consultation

## Abstract

The increasing prevalence of emergency room (ER) visits for mental health concerns presents a significant challenge for healthcare systems. This study aimed to analyze the socio-demographic and clinical characteristics associated with frequent users of psychiatric consultations in the ER of Maggiore della Carità University Hospital in Novara, Italy. A retrospective observational study was conducted over a six-year period (2017–2022), including all psychiatric consultations recorded in a hospital database. Frequent users were defined as individuals undergoing at least three psychiatric consultations in the ER within a year. Univariable and multivariable logistic models were employed to identify significant socio-demographic and clinical predictors of frequent use. Of the 1565 individuals who received psychiatric consultations in the ER, 92 (5.88%) were identified as frequent users. Factors associated with higher ER utilization included being unmarried (aOR 1.35, 95% CI 1.02–1.79), younger age (aOR 1.32, 95% CI 1.02–1.72), homelessness, diagnosis of schizophrenia, substance use disorder (aOR 1.49, 95% CI 1.06–2.09), and ongoing psychopharmacological treatment (aOR 1.56, 95% CI 1.12–2.18). These findings highlight the need for targeted interventions to improve care continuity and community-based support for individuals at risk of frequent ER visits for psychiatric reasons.

## 1. Introduction

Frequent utilization of Emergency Rooms (ERs) is a growing concern in healthcare, becoming more prevalent over time despite not always stemming from urgent health needs. While often perceived as a rapid response to various health and social issues, frequent ER visits are associated with poor patient outcomes, posing challenges for healthcare personnel, straining the system, and leading to escalating medical costs. Consequently, individuals who frequently use healthcare services are often referred to as ‘heavy-cost patients’ [1].

Variability regarding psychiatric ER utilization exists among nations, with healthcare system disparities influencing access and treatment patterns [2]. Hence, further details on psychiatric healthcare organization in Italy may be warranted. With the so-called Basaglia Law (Law 180/1978), Italy’s mental health system successfully transitioned from institutional to community-based care, which is meant to represent the core of psychiatric assistance and care. A key aspect of this reform was its emphasis on continuous and close outpatient follow-up, which proved effective in reducing psychiatric inpatient treatment and psychiatric-related ER visits. Barbato et al. (1998) [3] found that deinstitutionalization and the closing of psychiatric hospitals in Italy did not lead to an increase in compulsory psychiatric hospitalizations, suicides, or homelessness, suggesting that community-based care can effectively manage mental health conditions without overburdening emergency services. More recent data (from Angela Renata di Napoli et al. (2024) [4]) indicate that psychiatric-related admissions account for only 3.2% of all ER visits, highlighting the effectiveness of outpatient mental health services in preventing crises.

Nonetheless, addressing the issue of “frequent users” is essential to better understand how to manage this potentially critical issue. The definition of “frequent use” varies in the literature, ranging from ≥2 to ≥20 visits per year [5]. “Frequent users” of the ER typically make at least three to five visits within a year, representing a minority of ER patients but accumulating numerous visits and presenting complex needs, which may eventually contribute to adverse events, hospitalization, and increased mortality [6,7,8].

Various factors have been identified as correlates of frequent ER use, including high physical and mental health comorbidities, poor socio-economic status, substance abuse, and psychiatric disorders [9,10,11]. Patients with mental disorders and substance-related disorders significantly contribute to ER overcrowding, with consistent identification of a ‘mental health or alcohol/substance abuse’ profile among frequent ER users across different healthcare systems [12,13,14].

When examining frequent ER use for mental health reasons, studies suggest that financial and economic factors, socio-demographic factors, and clinical characteristics (such as age, homelessness, and specific diagnoses) play a key role [15]. Multiple studies have identified psychotic and depressive disorders, substance abuse, psychiatric service seeking outside office hours, and homelessness as major contributors to frequent ER use for mental health reasons, indicating the presence of distinct profiles among frequent users [16,17,18].

Given the evolving impact of the COVID-19 pandemic, including the lockdown phase, it is necessary to consider its influence on emergency psychiatric consultations. Although ER visits initially decreased during the lockdown, they significantly increased in the post-lockdown period [19]. Apart from quantitative changes, which can be explained by the lockdown, qualitative aspects must also be considered. For instance, suicide-related ER visits rose notably, likely driven by pandemic-induced social isolation and economic hardship [20]. Additionally, neuropsychiatric manifestations of COVID-19, including cognitive impairment and mood disturbances, contributed to increased ER consultations, with post-COVID syndrome emerging as a significant factor in psychiatric distress [21]. Furthermore, comorbid medical conditions, particularly in patients with severe psychiatric disorders such as schizophrenia and bipolar disorder, could further complicate emergency psychiatric care, increasing the burden on healthcare systems [22].

Frequent ER use for mental health reasons often involves patients with complex needs that extend beyond emergencies, encompassing behavioral health and social needs [23]. Therefore, a deeper understanding of this phenomenon and the trajectories of these patients is crucial. Implementing targeted strategies at both hospital and community levels, along with structured follow-up care, is essential to enhancing healthcare quality and efficiency [24].

This study was conducted in collaboration with the Psychiatry Unit and the Emergency Department of the Maggiore della Carità University Hospital in Novara, Italy, with the following objectives:To provide a descriptive analysis of the socio-demographic and clinical characteristics of ER “frequent users” (defined as individuals undergoing at least three psychiatric consultations within a year) for mental health-related reasons in the ER of the Maggiore della Carità University Hospital in Novara, Italy. For this objective, the socio-demographic and clinical features were those recorded at the first consultation in the study period.To compare the characteristics of consultations of ER ordinary service users (non- “frequent users”) with those of “frequent users” to identify potential differences between these two groups.To analyze, using univariable and multivariable logistic models, the most significant socio-demographic and clinical factors associated with frequent psychiatric consultations in the ER, considering each consultation as a discrete event.

## 2. Methods

This retrospective observational study analyzed psychiatric consultations conducted in the Emergency Room (ER) of the Maggiore della Carità University Hospital in Novara, Italy, over six years, from 1 January 2017 to 31 December 2022. “Frequent users” were defined as individuals with at least three ER visits requiring psychiatric consultation within 356.25 days. No exclusion criteria were applied; however, only patients aged 17 years or older were included, as those younger than 17 years are treated in a separate Pediatric ER, by institutional protocol.

To ensure reproducibility, a copy of the database variables, along with their respective encodings, will be made available upon request.

It is important to clarify that, in the following sections, the term ‘psychiatric consultations’ refers to the total number of ER psychiatric evaluations recorded in our dataset, as some patients underwent multiple consultations, resulting in a higher number of consultations than unique users. In contrast, ‘users’ refers to the individual patients who received at least one psychiatric consultation during the study period.

Socio-demographic and clinical data were retrospectively retrieved using the hospital’s specific software (PsNet, Hi.Tech SpA, Firenze, Italy). The variables analyzed included sex assigned at birth, age, residence (i.e., the city where the patient lives and the specific living situation—private residence, institutionalized setting, or homelessness), marital status, and educational status (classified as low if up to middle school and high if beyond middle school). Medical history was documented as a dichotomous variable (yes/no for organic comorbidities). Past psychiatric conditions included previous psychiatric diagnoses, prior psychiatric outpatient follow-ups, visits to the Outpatient Addiction Rehab Service (SERD), and past psychiatric hospitalizations. Current psychiatric conditions encompassed current psychiatric follow-up, SERD care, and psychopharmacological therapy. The DSM-5 classification [25] system was used to identify previous and/or current psychiatric conditions.

Further clinical data encompassed the type of ER access (voluntary vs. involuntary admissions, e.g., law enforcement referrals or psychiatric referrals), reason for consultation (main psychiatric symptomatology leading to ER access), appropriateness of consultation, assigned triage code (the urgency level assigned upon ER admission), psychiatric symptoms, and self-harming behavior, including suicidal intent. The classification of self-injurious behaviors included distinguishing whether they involved intent to die and the type of self-harm (e.g., self-cutting, precipitation or substance ingestion) ER management details, including psychopharmacological therapy and consultation outcomes (admission to the Psychiatry Unit, discharge from the ER, admission to other hospital wards, or scheduled outpatient care), were recorded. Additionally, any findings of alcohol or substance abuse in urine were documented.

Study data were collected and managed using REDCap (Research Electronic Data Capture) tools [26,27]; subject identities were encoded to identify potential multiple accesses, assigning a numerical code to each patient in a separate, protected dataset accessible only to authorized researchers. The study adhered to the RECORD guidelines, ensuring rigorous documentation of routinely collected health information.

To maintain confidentiality, all patient data were de-identified, and each subject was assigned a numerical code stored in a separate, secured dataset accessible only to authorized researchers.

The study received ethical approval from the Interhospital Territorial Ethics Committee of the Maggiore della Carità University Hospital in Novara (CE125/2023).

### Statistical Analysis

Each consultation was considered a discrete event, as described above. Descriptive statistical analyses were conducted separately for frequent users and non-frequent users. Frequencies were reported for categorical variables, while the means and standard deviations were reported for quantitative variables. Differences in proportions between groups were calculated using the Chi-square or Fisher test, and the differences in continuous variables between groups were tested using Student’s T-test after checking the normality assumption. Gender, age, marital status, and education were based on information at the first contact.

Univariable and multivariable logistic models were employed to evaluate the association between the characteristic of being frequent users of psychiatric consultations and socio-demographic variables (gender, age, marital status, and education), as well as those pertaining to current and previous psychiatric history (previous psychiatric issues, prior psychiatric hospitalizations, patient currently receiving psychiatric follow-up, past or current interactions with SERDs, presence of organic comorbidities, and undergoing psychopharmacological therapy). The stepwise selection method was used to compute the multivariable model. The multicollinearity between variables was checked before computing multivariable models using a correlation matrix, and all the correlation coefficients were lower than 0.5. The normality of residuals was checked using the Q-Q plot. The odds ratio and 95% confidence interval were calculated. The significance threshold was set at 0.05.

Statistical analyses were performed using SAS 9.4 and STATA version 17.

## 3. Results

### 3.1. Study Population

A total of 2399 psychiatric consultations involving 1565 individuals were retrieved from the PSNet database during the study period. On average, more than 350 consultations were conducted annually, with peaks exceeding 430 in 2019 and 2022. The number of “first” consultations, referring to patients with no previous contact with psychiatry in an emergency setting, exhibited an increasing trend over time.

Considering the whole study period (2017–2022), 1214 (77.57%) individuals had only one consultation during the study period, 200 (12.78%) had two consultations, 66 (4.22%) had three consultations, while the remaining 85 (5.43%) had more than three consultations.

Nonetheless, applying a criterion of fewer than 365 days between the first and third consultations, 92 individuals could be classified as frequent users, representing 5.88% of the total population of subjects receiving psychiatric consultations during the study period. Non-frequent users accounted for 1874 consultations, while frequent users contributed to 525 consultations.

### 3.2. Frequent Users and Non-Frequent Users: Univariable Logistic Models

The univariable logistic models showed several differences between frequent and non-frequent users of psychiatric consultations in the ER.

Demographic characteristics were compared between non-frequent users (*n* = 1473) and frequent users (*n* = 92), focusing on gender, marital status, and education at the first consultation during the study period. In terms of gender distribution, males constituted 50.0% of non-frequent users compared to 41.3% of frequent users, while females accounted for 50.0% and 58.7% of non-frequent and frequent users, respectively (OR = 1.42, CI 95%: 0.92–2.18 for females). As far as marital status is concerned, 35.0% of non-frequent users were married compared to 23.9% of frequent users, while non-married individuals were significantly more prevalent among frequent users (76.1% vs. 54.2%), (OR = 2.05, CI 95%: 1.25–3.36). Regarding education, a lower level of education (up to middle school) was observed in 49.3% of non-frequent users and 62.0% of frequent users; however, this difference was not statistically significant (OR = 0.93, CI 95%: 0.60–1.45). The mean age of non-frequent users was 46.6 years (SD = 18.9), compared to 43.7 years (SD = 14.1) for frequent users (*p*-value = 0.15). Overall, these results suggest that frequent users were more likely to be unmarried and tended to have lower educational attainment (see Table 1).

Significant differences in psychiatric and medical histories were observed between non-frequent users and frequent users. A markedly higher proportion of frequent users had a past psychiatric condition (95.4% vs. 63.5%, OR = 13.3, CI 95%: 8.40–20.97) and had previously experienced psychiatric hospitalizations (75.4% vs. 24.8%, OR = 7.36, CI 95%: 5.80–9.32). Current psychiatric follow-up was also markedly more prevalent among frequent users (80.9% vs. 48.0%, OR = 4.61, CI 95%: 3.62–5.87). While a greater proportion of frequent users had a SERD history (17.0% vs. 10.3%, OR = 1.75, CI 95%: 1.33–2.31), the significance of this difference was less pronounced. Additionally, current psychopharmacological therapy was reported by 85.3% of frequent users compared to 61.5% of non-frequent users (OR = 3.78, CI 95%: 2.90–4.92). The prevalence of comorbid somatic conditions was slightly higher in frequent users (43.4% vs. 38.6%, OR = 1.20, CI 95%: 0.98–1.46). However, current SERD follow-up showed minimal differences between the groups (7.2% vs. 5.4%, OR = 1.35, CI 95%: 0.92–2.00) (see Table 2).

Furthermore, the analysis indicated several significant differences in current symptomatology between the two groups. Notably, psychomotor agitation was significantly more frequent among frequent users (22.9%) compared to non-frequent users (15.2%) (*p*-value < 0.0001). Similarly, mood alterations were more prevalent among non-frequent users (12.8%) than frequent users (9.5%), *p* = 0.04. Cognitive alterations (including symptoms such as memory impairment, disorientation, and executive function difficulties) and delirium (characterized by acute confusion, fluctuating consciousness levels, and attention deficits) were significantly less prevalent among frequent users (3.2% vs. 5.9%, *p* = 0.01). Additionally, other neurological symptoms, including extrapyramidal symptoms, were less frequent among frequent users (2.1% vs. 4.5%, *p* = 0.01).

Self-injury behaviors were reported by 17.3% of non-frequent users compared to 13.3% of frequent users (*p* = 0.03). Regarding the type of self-injury, no significant differences were observed between the two groups for drug ingestion/overdosing or self-cutting injuries (*p*-values 0.24 and 0.35, respectively). Additionally, among those who ingested drugs, no significant differences emerged concerning the type of substance used., However, the ingestion of SSRIs and other antidepressants was slightly more common among frequent users (20.5% vs. 11.0%, *p* = 0.09).

Regarding medication used in the context of the ER access, the administration of benzodiazepines was similar between both groups (61.8% in non-frequent users and 63.0% in frequent users, *p* = 0.73). Additionally, no statistically significant differences were found in the use of antipsychotics and combinations of medications between the two groups (see Table 3).

### 3.3. Multivariable Logistic Regression

We examined the socio-demographic and clinical variables associated with the likelihood of being a frequent user; the adjusted odds ratios (aOR) and corresponding 95% confidence intervals (CI) are reported in Table 4. Females were 1.40 times more likely than males to be frequent users (CI: 1.09–1.80). Non-married individuals had an increased likelihood of 1.35 times of being frequent users, compared to married ones (CI: 1.02–1.79). Individuals with a past psychiatric condition and those with previous contact with the SERD were 3.87 (CI: 2.28–6.59) and 1.49 (CI: 1.06–2.09) times more likely to be frequent users, respectively. A previous psychiatric hospitalization was strongly associated with being a frequent user (aOR 4.02; CI: 3.05–5.32). Those currently taking psychopharmacological therapy had a 1.56 times higher likelihood of experiencing the outcome compared to those not in therapy (CI: 1.12–2.18) (see Table 4). Furthermore, the presence of somatic comorbidities increased the likelihood of frequent psychiatric consultations in the ER by 1.37 times (CI: 1.07–1.77).

## 4. Discussion

This study aimed to analyze the socio-demographic and clinical characteristics of ER frequent users, compare them to non-frequent users, and identify the potential correlates of frequent ER utilization for psychiatric concerns. There is an urgent need for a more comprehensive understanding of frequent ER utilization for mental health issues in Italy, which could be crucial both for clinical management and healthcare policy.

### 4.1. Descriptive Analysis: Study Population

The number of psychiatric consultations at the ER of our University Hospital fluctuated over the study period. On average, more than 350 consultations were conducted annually, with a notable increase in 2022 (up to 450 consultations). This trend is likely related to the COVID-19 pandemic, which initially led to a decrease in consultations during the lockdown period, followed by changes in healthcare-seeking behavior. This observation aligns with findings from the Mental Health Report 2020 by the Ministry of Health [28].

A total of ninety-two patients, accounting for 5.88% of all psychiatric consultations, were identified as frequent users throughout the study period. This prevalence is consistent with existing literature [9,11]. Nonetheless, it should be underscored that the comparison of the current findings with those reported by other studies is challenging due to variations in definitions of frequent users, clinical settings, healthcare systems, and cultural backgrounds [29].

### 4.2. Comparison Between Consultations of Frequent Users and Non-Frequent Users: Univariable Logistic Model

Our findings indicate that frequent users are more likely to be unmarried, while no statistically significant differences were found in sex assigned at birth, age, or education level. These findings are partially consistent with similar studies: indeed, these report a higher prevalence of male subjects among frequent users [12]. However, the observation that frequent users are often single or not married aligns with existing literature [30,31]. It can be hypothesized that stable relationships, such as marriage, may provide more significant support in case of crises, and reduce the likelihood of seeking help in the ER in difficult situations.

Frequent users accessed the ER more frequently due to psychiatric or social/legal motivations, a pattern consistent with previous research [16]. Individuals without psychiatric needs may be brought to the ER by law enforcement for public order issues, leading to psychiatric consultations. Similarly, individuals without a fixed residence may seek shelter in the ER [32,33]. This issue warrants attention from healthcare policymakers, as it highlights a broader misunderstanding of the roles of both the ER and psychiatric services, which may be utilized beyond their intended scope. Additionally, limitations in infrastructure and social support systems may play a significant role in driving individuals to seek psychiatric services in the ER, surpassing its intended use.

In this study, frequent users were more likely to access the ER voluntarily or upon referral from a psychiatrist or SERD physician. The findings suggest that these patients often have a past psychiatric condition, previous hospitalizations, or contacts with SERDS, indicating complex care needs. They were also more likely to be receiving psychopharmacological therapy, to have a current or previous psychiatric diagnosis or prescription by another physician, or, in some cases, to be self-medicating. Furthermore, frequent users primarily sought ER consultations for anxious symptoms or psychomotor agitation [34,35,36], suggesting potential challenges in the integrated management of these disorders by the National Health System [35,37,38,39]. On the other hand, no statistically significant differences were found between frequent and non-frequent users regarding alcohol and substance abuse testing, which contradicts previous studies indicating a higher prevalence of substance-related issues among frequent ER users [31,40,41]. This discrepancy may be attributed to differences in study populations, settings, or methodologies.

The study also highlighted the issue of substance use in the Novara territory, consistent with the European Drug Report’s findings on the growing availability of illicit substances. Despite this, robust collaboration between Psychiatry and the ER has facilitated standardized substance abuse screening, contributing to a decline in psychiatric consultations for intoxication-related cases. This contrasts with previous studies that have reported a high demand for psychiatric consultations due to alcohol or substance intoxications among frequent ER users [16,31].

No statistically significant differences emerged between frequent and non-frequent users concerning psychotic disorders, mood disorders, or self-harm, in contrast to existing literature. This finding suggests that patients with these disorders may receive more effective crisis management in outpatient settings, thereby reducing the necessity for multiple ER visits. From a healthcare management perspective, this interpretation is encouraging, particularly considering the reforms introduced by the so-called Basaglia Law, which aimed to strengthen mental health care at the community level.

Hospitalization in a psychiatric ward was significantly more common among frequent users, consistent with previous research [42,43]. This finding suggests that frequent users may have more severe psychiatric conditions requiring hospitalization or experience crises that are difficult to manage, leading psychiatrists to the choice of inpatient admission. This trend highlights ongoing challenges in transitioning from hospital-centric care to a community-based mental health system. Furthermore, frequent hospitalizations impose economic burdens on the National Health System, highlighting the need for improved resource management and alternative strategies to optimize mental health care.

Nonetheless, the results of the univariable logistic model should be interpreted with caution, as they provide a lower level of evidence compared to the multivariable logistic model, particularly given the study design, which did not include control groups. While univariable analyses offer useful preliminary insights, they do not account for potential confounding factors that may influence the observed associations.

In contrast, the multivariable logistic model allows for a more robust analysis by simultaneously adjusting for multiple variables. This approach better reflects the complexity of frequent psychiatric ER use and provides stronger evidence for the factors independently associated with this phenomenon. Notably, our multivariable analysis confirmed that being female, unmarried, having a prior psychiatric history, undergoing psychopharmacological therapy, and having past interactions with SERDs were significantly associated with frequent ER psychiatric consultations. These findings align with existing literature and emphasize the need for targeted interventions addressing the specific vulnerabilities of this population.

Future research should prioritize multivariable analyses to refine our understanding of the interplay between socio-demographic and clinical factors, as well as to guide more effective preventive and management strategies.

### 4.3. Multivariable Logistic Model

The multivariable logistic regression model allowed the identification of a distinct profile of frequent users, which was characterized by several key demographic and clinical factors. Specifically, this profile includes individuals designated as female at birth, those who are unmarried, and individuals with a documented history of psychiatric disorders. Additionally, the analysis revealed that frequent users were more likely to be currently taking psychopharmacological medication and to have had prior interactions with Outpatient Rehabilitation Services. These findings are consistent with the existing literature that underscores the importance of these factors in understanding the characteristics of frequent users [44,45,46]. By delineating this profile, the results should foster the development of targeted interventions aimed specifically at this subgroup, thereby enhancing the effectiveness of treatment strategies and resource allocation. The findings suggest that policy efforts should focus on early intervention, crisis management alternatives, and enhanced coordination between psychiatric and social services to reduce unnecessary ER visits. Future research should explore longitudinal trends, social determinants, and alternative care models to refine strategies for managing frequent users more effectively. Such targeted approaches could potentially improve clinical outcomes and promote better management of their unique needs.

### 4.4. Strengths and Limitations

We decided to focus our analyses on frequent users, but considering each of the consultations received in the ER setting as a discrete event. As we wanted to assess the phenomenon from the perspective of the ER physician and of the consultant psychiatrist in the ER setting, we believed this was the best approach. Furthermore, as the study period included 6 years, some variables (marital status, living accommodation, employment, being treated by a health service, etc.) could possibly change, but given the design of the study, which is neither longitudinal nor a follow-up, we decided to address this issue assessing the features of patients at each consultation. Of course, this choice can possibly lead to some biases, inflating some features, which are counted more times for frequent users, but as some of these features are constant while other are more volatile, the other approach (considering individuals and not consultations) would also have led to biases, as some patients would possibly have been observed in one year but not in another one.

Nonetheless, we believe that this study, conducted over a six-year period (2017–2022), provided valuable insights into the phenomenon of frequent users of psychiatric consultations in the ER setting, contributing to the limited literature on this topic in Italy. Multivariable analyses identified specific characteristics of this patient subgroup, which could help inform targeted interventions.

However, the study’s single-center nature at Maggiore della Carità University Hospital in Novara, while representative of the Piedmont region, limits its generalizability. Future research should involve multiple centers to increase the sample size and enhance the representativeness of findings, an effort that is currently underway. Additionally, further investigation could focus on more specific subgroups of frequent users, categorized based on diagnostic or socio-demographic characteristics, to better understand their unmet needs, as suggested in previous studies [41].

## 5. Conclusions

Deinstitutionalization policies in psychiatry have posed significant challenges in addressing both acute psychiatric crises and the long-term management of psychiatric conditions. Despite its acknowledged limitations, our study aligns with these considerations.

Nonetheless, the results of the univariable logistic model should be interpreted with caution, as they provide a lower level of evidence compared to the multivariable logistic model, particularly given the study design, which did not include control groups. While univariable analyses offer useful preliminary insights, they do not account for potential confounding factors that may influence the observed associations.

In contrast, the multivariable logistic model allows for a more robust analysis by simultaneously adjusting for multiple variables. This approach better reflects the complexity of frequent psychiatric ER use and provides stronger evidence for the factors independently associated with this phenomenon. Notably, our multivariable analysis confirmed that being female, unmarried, having a prior psychiatric history, undergoing psychopharmacological therapy, and having past interactions with SERDs were significantly associated with frequent ER psychiatric consultations. These findings align with the existing literature and emphasize the need for targeted interventions addressing the specific vulnerabilities of this population.

Recent studies [13] have focused on intervention strategies to reduce frequent emergency department visits. Enhanced care continuity across territorial services and fostering collaboration among healthcare professionals has proven effective. Prioritizing the therapeutic alliance between caregivers and patients, along with individualized care models, may reduce visits, decrease stigma, and improve treatment adherence [47,48]. Enhancing outpatient and community-based psychiatric care, improving coordination between psychiatric and social services, expanding alternative crisis interventions, and implementing early intervention programs are essential strategies to reduce emergency department visits and hospitalizations among frequent psychiatric service users.

Future research should prioritize multivariable analyses to refine our understanding of the interplay between socio-demographic and clinical factors, as well as to guide more effective preventive and management strategies. Understanding the complexities of frequent ER users is crucial from multiple perspectives. This issue affects both patients, who often do not receive appropriate responses to their needs in emergency settings, and healthcare professionals—whether emergency physicians or psychiatrists—who may struggle to adequately address requests for assistance that often extend beyond strictly medical concerns. This mismatch between patient needs and available resources may contribute to frustration on both sides.

## Figures and Tables

**Table 1 ijerph-22-00828-t001:** Comparison between frequent users and non-frequent users: socio-demographic variables as recorded at the first consultation during the study period.

	Non-Frequent Users (*N* = 1473) *n* (%)	Frequent Users (*N* = 92) *n* (%)	OR (CI95%)
Gender			
Male	736 (50.0)	38 (41.3)	1
Female	737 (50.0)	54 (58.7)	1.42 (0.92–2.18)
Marital status			
Married	515 (35.0)	22 (23.9)	1
Not married	798 (54.2)	70 (76.1)	2.05 (1.25–3.36)
Missing	160 (10.9)	--	
Education			
Low level (up to middle school)	726 (49.3)	57 (62.0)	1
High level	477 (32.4)	35 (38.0)	0.93 (0.60–1.45)
Missing	270 (18.3)	--	
			*p*-value
Age, years Mean (SD)	46.6 (18.9)	43.7 (14.1)	0.15

Note: Data are presented as absolute numbers (*n*) and percentages (%). OR = Odds Ratio; CI = Confidence Interval. The reference category for gender is male, and for marital status, it is married. The total number of patients included in this analysis is 1565, with 92 frequent users and 1473 non-frequent users. Missing data were not included in the calculations.

**Table 2 ijerph-22-00828-t002:** Comparison of variables regarding remote psychiatric history and organic comorbidities.

	Consultations of Non-Frequent Users (N = 1874) *n* (%)	Consultations of Frequent Users (N = 525) *n* (%)	OR (CI95%)
Past psychiatric condition	1189 (63.5)	501 (95.4)	13.3 (8.40–20.97)
SERD history	193 (10.3)	89 (17.0)	1.75 (1.33–2.31)
Previous psychiatric hospitalizations	365 (24.8)	396 (75.4)	7.36 (5.80–9.32)
Current psychiatric follow-up	900 (48.0)	425 (80.9)	4.61 (3.62–5.87)
Current SERD follow-up	101 (5.4)	38 (7.2)	1.35 (0.92–2.00)
Organic comorbidities	724 (38.6)	228 (43.4)	1.20 (0.98–1.46)
Current psychopharmacological therapy	1153 (61.5)	448 (85.3)	3.78 (2.90–4.92)

Note: Data are presented as absolute numbers (*n*) and percentages (%). OR = Odds Ratio; CI = Confidence Interval. Past psychiatric condition includes any previous psychiatric diagnosis, treatment, or hospitalization. SERD = Outpatient Addiction Rehabilitation Service. The reference category for each variable is the absence of the characteristic (e.g., no past psychiatric condition, no previous psychiatric hospitalization, no organic comorbidities).

**Table 3 ijerph-22-00828-t003:** Comparison of variables regarding psychiatric proximate history and self-injurious gestures.

	Consultations of Non-Frequent Users (N = 1874) *n* (%)	Consultations of Frequent Users (N = 525) *n* (%)	*p* Value
Main symptomatology			
Anxious state	574 (30.6)	175 (33.3)	0.24
Psychomotor agitation	284 (15.2)	120 (22.9)	<0.0001
Psychotic symptomatology	258 (13.8)	75 (14.3)	0.77
Mood alteration	239 (12.8)	50 (9.5)	0.04
Intoxication or withdrawal	152 (8.1)	37 (7.0)	0.41
Negative psychic examination	165 (8.8)	36 (6.9)	0.16
Cognitive alterations	111 (5.9)	17 (3.2)	0.01
Other (extrapyramidal symptoms, neurological symptoms)	85 (4.5)	11 (2.1)	0.01
Self-injurious gesture	324 (17.3)	70 (13.3)	0.03
Anti-conservative intent	144 (7.7)	42 (8.0)	0.82
Act type (in subjects with acting out)			
Drug ingestion	190 (10.1)	44 (8.4)	0.24
Cutting injuries	62 (3.3)	13 (2.5)	0.35
Other (e.g., caustics, CO)	70 (3.7)	12 (2.3)	0.12
Type of ingestion (in subjects with ingestion)			
Benzodiazepines or barbiturates	65 (34.2)	13 (30.0)	0.59
SSRI or other antidepressants	21 (11.0)	9 (20.5)	0.09
Non-psychiatric drugs	30 (15.8)	6 (13.6)	0.71
Various drug intake	53 (27.9)	14 (31.8)	0.61
Antipsychotics	8 (4.2)	1 (2.3)	0.59
Other	11 (5.8)	1 (2.3)	0.34
Psychopharmacological therapy administered in the ER			
Benzodiazepines	549 (61.8)	150 (63.0)	0.73
Antipsychotics	171 (19.2)	41 (17.2)	0.48
Combination of benzodiazepines and antipsychotics	139 (15.6)	44 (18.5)	0.28
Other (e.g., biperidene or metadoxine)	30 (3.4)	3 (1.3)	0.09

Note: Data are presented as absolute numbers (*n*) and percentages (%). Self-injurious behaviors include suicide attempts and other acts of self-harm. The reference category for the symptomatology variables is the absence of the specific symptom. Data on psychopharmacological therapy refer to medications administered during the ER consultation.

**Table 4 ijerph-22-00828-t004:** Multivariable logistic model.

	aOR (CI95%)
Gender (female vs. male)	1.40 (1.09–1.80)
Marital status (not married vs. married)	1.35 (1.02–1.79)
Age (<50 vs. ≥50)	1.32 (1.02–1.72)
Psychiatric history (yes vs. no)	3.87 (2.28–6.59)
SERD history (yes vs. no)	1.49 (1.06–2.09)
Previous psychiatric hospitalizations (yes vs. no)	4.02 (3.05–5.32)
Organic comorbidities (yes vs. no)	1.37 (1.07–1.77)
Current psychopharmacological therapy (yes vs. no)	1.56 (1.12–2.18)

Note: aOR = Adjusted Odds Ratio; CI = 95% Confidence Interval. The multivariable model includes socio-demographic and clinical factors identified as significant in univariable analyses. The reference categories are male (gender), married (marital status), age ≥ 50 years, no psychiatric history, no previous psychiatric hospitalization, no SERD history, no organic comorbidities, and no current psychopharmacological therapy.

## Data Availability

The data presented in this study are available on request from the corresponding author due to privacy rules.

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
