# Peer review of "Frequent Users in Psychiatric Consultations: A 6-Year Retrospective Study in the Emergency Department"

_ijerph, 2025, doi:10.3390/ijerph22060828_

Round 1
Reviewer 1 Report
Comments and Suggestions for Authors
dear authors.
It is a good article, I suggest only minor corrections to improve the redaction. I think you must improve the tables, specifically table 2, and table 3. In these tables, it appears that you are comparing consultations rather than users, and this will be a bias into the analysis. For example: If you are comparing consultations, comorbidity rates may appear inflated because a frequent user (three ER visits) with a comorbidities could be counted three times.
I send out comments in the attachment file.

Author Response
It is a good article, I suggest only minor corrections to improve the redaction. I think you must
improve the tables, specifically table 2, and table 3. In these tables, it appears that you are
comparing consultations rather than users, and this will be a bias into the analysis. For example: If
you are comparing consultations, comorbidity rates may appear inflated because a frequent user
(three ER visits) with a comorbidities could be counted three times.
We have added a comment in the limitation section concerning this. Anyway, there are some
reasons why we chose to compare consultations (and patients’ features at each consultations)
instead of individuals receiving the consultations, which we have described in the text, including
the following:
1. The approach of counting independently each visit, considering it a discrete event, is not
uncommon in similar studies;
2. We wanted to focus more on “what happened” in the specific consultation, than on individuals,
hence we believed this was the best approach;
3. As the study period included 6 years, some variables (marital status, living accommodation,
employment, being treated by a health service…) could possibly change, and as our study is
not a follow-up longitudinal study, it would be impossible to address these changes in a proper
way. Also, as the accesses of each patient during the study period may have been very uneven
(e.g. some may have had 3 or more accesses in the first year of study and then none, other
may have had one access a year…), our approach seemed the best one.
Comments within the text
We have removed the bracket before (Furthermore, on page 2, thank you for noticing the typo
Page 3: you suggested replacing “intent” with “attempt” but actually we have differentiated suicide
attempts from the intention to die, therefore we believe it is more correct not to replace the word
“intent”.
On page 4 we have rephrased in order to make it clearer the percentage of frequent users
according to the definition we used, which is not equal to the sum of those with 3 or more
consultations during the whole 6-year period of the study. We hope that now it is easier to read and
understand.
We have checked the use of the acronyms in the text.
Reviewer 2 Report
Comments and Suggestions for Authors
This is a retrospective study undertaken over six years; however, the variables that form part of the findings are volatile, suggesting that they have the potential to affect the outcome if revisited. The study remains silent on this. Please refer to the file as attached.

Author Response
This is a retrospective study undertaken over six years; however, the variables that form part of the
findings are volatile, suggesting that they have the potential to affect the outcome if revisited. The
study remains silent on this. Please refer to the file as attached.
Factors such as younger age and homelessness might change, at the end of the study period,
where
they revisited?
The study hypothesises the fact that marriage and stable relationships reduces the likelihood of
accessing ER through support, yet, in the 6-year period there is no mention of how many
participants of 1565’s had a status change, either through marriage or having stable relationships?
1. Where variables such as marital and educational statuses of all 1565 participants remained the
same in the final/6th year of the study?
2. If there was a change, how did the change affect the behaviour?
3. Does the 92 participants who frequented ERs’ status at the start and the end of the study period
remained the same? Such as moving away from a stable relationship or acquiring education?
4. There is silence in the review of these volatile variables of the 92 frequent users, and their
potential to influence utilisation of the ER
Unmarried status might change, how was this affected at the start to the end of the study?
You are certainly right; we have added a comment about this in the limitation section of the study
concerning this. Anyway, there are some reasons why we chose to compare consultations (and
patients’ features at each consultations) instead of individuals receiving the consultations, which
we have described in the text, including the following:
1. The approach of counting independently each visit, considering it a discrete event, is not
uncommon in similar studies;
2. We wanted to focus more on “what happened” in the specific consultation, than on individuals,
hence we believed this was the best approach;
3. As the study period included 6 years, some variables (marital status, living accommodation,
employment, being treated by a health service…) could possibly change, and as our study is
not a follow-up longitudinal study, it would be impossible to address these changes in a proper
way. Also, as the accesses of each patient during the study period may have been very uneven
(e.g. some may have had 3 or more accesses in the first year of study and then none, other
may have had one access a year…), our approach seemed the best one.